# Enhancing the Mechanical Properties of Corn Starch Films for Sustainable Food Packaging by Optimizing Enzymatic Hydrolysis

**DOI:** 10.3390/polym15081899

**Published:** 2023-04-15

**Authors:** Andra-Ionela Ghizdareanu, Alexandra Banu, Diana Pasarin, Andreea Ionita (Afilipoaei), Cristian-Andi Nicolae, Augusta Raluca Gabor, Delia Pătroi

**Affiliations:** 1Faculty of Material Science and Engineering, University Politehnica of Bucharest, 313 Splaiul Independentei, 060042 Bucharest, Romania; 2National Research and Development Institute for Chemistry and Petrochemistry-ICECHIM, 202 Splaiul Independentei, 060021 Bucharest, Romania; 3National Institute for Research and Development in Electrical Engineering, ICPE-CA, 313 Splaiul Unirii, 030138 Bucharest, Romania

**Keywords:** Box–Behnken design, enzymatic hydrolysis, corn starch, mechanical properties, packaging films

## Abstract

The objective of this study was to investigate the effects of enzymatic hydrolysis using α-amylase from *Bacillus amyloliquefaciens* on the mechanical properties of starch-based films. The process parameters of enzymatic hydrolysis and the degree of hydrolysis (DH) were optimized using a Box–Behnken design (BBD) and response surface methodology (RSM). The mechanical properties of the resulting hydrolyzed corn starch films (tensile strain at break, tensile stress at break, and Young’s modulus) were evaluated. The results showed that the optimum DH for hydrolyzed corn starch films to achieve improved mechanical properties of the film-forming solutions was achieved at a corn starch to water ratio of 1:2.8, an enzyme to substrate ratio of 357 U/g, and an incubation temperature of 48 °C. Under the optimized conditions, the hydrolyzed corn starch film had a higher water absorption index of 2.32 ± 0.112% compared to the native corn starch film (control) of 0.81 ± 0.352%. The hydrolyzed corn starch films were more transparent than the control sample, with a light transmission of 78.5 ± 0.121% per mm. Fourier-transformed infrared spectroscopy (FTIR) analysis showed that the enzymatically hydrolyzed corn starch films had a more compact and solid structure in terms of molecular bonds, and the contact angle was also higher, at 79.21 ± 0.171° for this sample. The control sample had a higher melting point than the hydrolyzed corn starch film, as indicated by the significant difference in the temperature of the first endothermic event between the two films. The atomic force microscopy (AFM) characterization of the hydrolyzed corn starch film showed intermediate surface roughness. A comparison of the data from the two samples showed that the hydrolyzed corn starch film had better mechanical properties than the control sample, with a greater change in the storage modulus over a wider temperature range and higher values for the loss modulus and tan delta, indicating that the hydrolyzed corn starch film had better energy dissipation properties, as shown by thermal analysis. The improved mechanical properties of the resulting film of hydrolyzed corn starch were attributed to the enzymatic hydrolysis process, which breaks the starch molecules into smaller units, resulting in increased chain flexibility, improved film-forming ability, and stronger intermolecular bonds.

## 1. Introduction

Synthetic plastics have dominated every aspect of human activity, especially in the packaging sector, and despite their many benefits they have long been a major environmental concern. The increasing use of these plastics has led to severe energy crises due to their non-biodegradability and dependence on a non-renewable petroleum resource, as well as pollution from their disposal, which has damaged the environment, water sources, sewage systems, rivers, and streams [1]. Starchy plants such as potatoes, corn, tubers, cassava, and cornmeal can be used to make an environmentally friendly plastic known as bioplastic [2]. Natural polymers are attracting a lot of attention because they can serve as a source for making environmentally friendly, usable products to replace those made from petroleum. Green composites have also become increasingly popular in recent years, because they offer several advantages over conventional composites. These composites are made from natural fibers such as seed fibers, grass fibers, leaf fibers, bast fibers, and others. Let us take a closer look at some of the most commonly used natural fibers in green composites. Seed fibers, such as coir and kapok, are lightweight and have good thermal insulation properties. They are often used in the construction industry for insulation and soundproofing. Grass fibers, including straw, are inexpensive and readily available [3]. They are used to make environmentally friendly materials such as biodegradable packaging and paper products. Leaf fibers, such as sisal and agave, are strong and durable, making them suitable for a variety of applications including auto parts, furniture, and sports equipment. Bast fibers such as hemp, jute, and flax are commonly used to make green composites. Hemp fibers, for example, have high tensile strength and are resistant to ultraviolet light and moisture, making them ideal for producing biodegradable plastic composites [4]. Starch is preferred for this purpose because it is naturally biodegradable, renewable, affordable, and can be obtained from many plants [5]. Starch is a complex carbohydrate that is often used as a raw material for various industrial applications, including the production of films. When starch is hydrolyzed using enzymes such as amylase, its structure and functionality change significantly, which can improve the quality of the resulting film [6]. Structurally, starch is composed of two types of glucose polymers: amylose and amylopectin. Amylose is a linear polymer, while amylopectin is a branched polymer with many side chains. During enzymatic hydrolysis, enzymes break down the long chains of amylose and amylopectin into smaller glucose molecules. This process results in a decrease in the molecular weight of starch and an increase in the number of free glucose molecules in the solution. Alpha-amylase, also called dextrinogenic amylase, causes the non-selective, random endo hydrolysis of α-(1,4) glycosidic bonds in amylose and amylopectin. This amylase produces maltose, maltotriose, and higher oligosaccharides from amylose, as well as maltose and glucose, and also limit dextrins from amylopectin [7]. Alpha-amylases can be divided into (i) maltogenic amylase, which produces maltose by cleaving the bonds at the nonreducing ends of amylose molecules, (ii) maltotriohydrolase, which cleaves the bonds at the non-reducing ends to produce maltotriose, (iii) maltotetrahydrolase, which cleaves the same bonds to produce maltotetraose, (iv) a-maltohexahydrolase, which produces maltohexaose, and so on. These variants differ from each other in the type of attack, which can be either random multichain or single-chain multiple [8]. Amylose is more readily hydrolyzed than amylopectin, and the ratio of the rate of hydrolysis of amylose to amylopectin can reach 2:1 [9]. For the hydrolysis of amylopectin, alpha-amylase is recommended as the enzyme that allows the nonrandom cleavage of the molecule. Models with a predictive value for the results of saccharification of starches have been developed based on their structure and, in particular, on the pattern of their branching. However, these models are not uniformly valid, since no relationship has been found between starch constitution and the results of starch hydrolysis with amylase performed under different conditions. From a functional point of view, the hydrolysis of starch leads to changes in its rheological properties, such as viscosity and gelatinization temperature. Hydrolyzed starch has a lower viscosity than unhydrolyzed starch, which makes it easier to process into films. In addition, hydrolyzed starch has a lower gelatinization temperature, which means it can be processed at lower temperatures, reducing the risk of thermal degradation. Enzymatic hydrolysis can improve the film-forming properties of starch by increasing its solubility and reducing its tendency to retrograde, i.e., return to its crystalline form [10]. This results in a more uniform and transparent film with improved mechanical properties, such as higher tensile strength and elongation. Carbohydrate-based films often have higher mechanical strength and can serve as gas barriers. Enzymatic hydrolysis uses enzymes to break down complex molecules, such as starch, into smaller, more soluble units. Corn starch is a commonly used feedstock for enzymatic hydrolysis because it is abundant, inexpensive, and renewable. The hydrolyzed corn starch thus obtained can be used as a raw material for various applications, including the production of biodegradable packaging films [11].

Films of hydrolyzed corn starch can be prepared by casting the hydrolyzed starch solution into thin films, which are then dried to remove excess water. The films can be further modified by adding plasticizers, crosslinking agents, or other additives to improve their mechanical properties, barrier properties, and biodegradability [12]. The use of hydrolyzed corn starch films as a sustainable packaging material offers several advantages over traditional petroleum-based plastics, including biodegradability, reusability, and reduced environmental impact [13]. However, producing films from hydrolyzed corn starch can be challenging due to their poor mechanical properties and water sensitivity. To overcome these challenges, ongoing research and development efforts are focused on optimizing the enzymatic hydrolysis process, exploring new processing techniques, and developing new formulations and additives to improve the performance of hydrolyzed corn starch films [14].

The enzymatic hydrolysis of corn starch is an important process for the production of biodegradable packaging films. The enzymatic hydrolysis of corn starch breaks down the long chains of glucose molecules into smaller, water-soluble molecules, which are then more easily combined with glycerol to form a biodegradable film. The enzymatic hydrolysis process not only makes the corn starch more soluble and easier to process, but also improves the biodegradability of the resulting films. This is because the smaller glucose molecules are more accessible to soil microorganisms, which can degrade the film much more quickly once it is disposed of in nature [15]. The optimization of this process can lead to films with desired properties such as high mechanical strength, good barrier properties, and biodegradability [16]. Three key parameters that can be optimized for the enzymatic hydrolysis of corn starch are the ratio of corn starch to water, the ratio of enzyme concentration to the substrate, and the temperature of incubation. The ratio of corn starch to water affects the concentration of corn starch in the reaction mixture, and thus the efficiency of enzymatic hydrolysis. A higher concentration of corn starch can result in a lower efficiency of enzymatic hydrolysis due to the increased viscosity and decreased accessibility of the enzyme to the starch molecules. However, a lower concentration of corn starch may result in a lower yield of the final product. The optimum ratio of corn starch to water depends on the enzyme used and the desired properties of the end product [17]. The ratio of enzyme concentration to substrate determines the amount of enzyme added to the reaction mixture relative to the amount of substrate available for hydrolysis. A higher enzyme concentration can lead to a higher efficiency of the hydrolysis reaction and a higher yield of the final product. However, a high enzyme concentration can also lead to the inactivation of the enzyme and a higher cost for the process. One reason for the inactivation of enzymes at high concentrations is the phenomenon known as autoinhibition, which occurs when enzymes bind to their own substrate at high concentrations, resulting in a decrease in enzymatic activity. In addition, the high enzyme concentration can also lead to non-specific binding and aggregation, which can further decrease enzyme activity and cause higher process costs [18]. The optimal ratio of enzyme concentration to substrate depends on the enzyme used [19]. The temperature of incubation affects the rate of the hydrolysis reaction, and thus the efficiency of enzymatic hydrolysis. A higher temperature can increase the reaction rate and lead to a higher yield of the final product. However, high temperatures can lead to the denaturation of the enzyme and a lower yield of the final product. The optimal temperature for incubation depends on the enzyme used and the desired properties of the final product. Several studies have reported on the optimization of these parameters for the enzymatic hydrolysis of corn starch [20,21]. For example, Chen L et al. optimized the ratio of corn starch to water, enzyme concentration to the substrate, and the temperature of incubation using response surface methodology and reported improved yield and properties of the final product [22]. Similarly, Luo S et al. optimized the same parameters using the BBD and reported improved mechanical properties and higher water vapor permeability of the final product [23].

Optimizing the ratio of corn starch to water, enzyme concentration to the substrate, and incubation temperature is important for the efficient enzymatic hydrolysis of corn starch and the production of high-quality biodegradable packaging films. To develop a model for the mechanical properties of hydrolyzed starch-based films as a function of composition after hydrolysis or as a function of DH, an optimization process can be carried out using BBD and the RSM. Previous studies have mostly focused on the development of hydrolyzed starch-based films, but none have specifically focused on creating a model for the mechanical properties of films based on the enzymatic hydrolysis parameters of starch. To achieve the ideal DH and mechanical properties for films made from hydrolyzed corn starch, the process of enzymatic hydrolysis must be monitored and optimized by variations in the addition of the ratio of corn starch to water, the enzyme, and the different temperatures of hydrolysis activity. This is important because improved mechanical properties are needed to evaluate the strength of bioplastics to external loads, with tensile strength being the greatest force bioplastics can withstand, which is influenced by the addition of plastic material [24]. The BBD and the RSM are often used to optimize experiments with bioplastics. The BBD is a statistical experimental design that considers all possible combinations of certain dependent variables to analyze the effects of multiple variables on a response variable. The response surface method, on the other hand, is a mathematical and statistical modeling technique for observing, predicting, and improving a response that is influenced by several independent variables [25,26].

This study aimed to investigate the influence of DH on the mechanical properties of starch-based films by optimizing enzymatic hydrolysis. The BBD and RSM were used to develop models and investigate the individual and interactive effects of the process parameters of enzymatic hydrolysis and DH on the mechanical properties (tensile strain at break, tensile stress at break, and Young’s modulus) of hydrolyzed corn starch films.

## 2. Materials and Methods

SCM Colin Daily Romania supplied the corn starch, while SC Chimreactiv SRL provided the glacial acetic acid and glycerol. Sigma-Aldrich supplied the α-amylase *Bacillus amyloliquefaciens*, with an enzyme activity of >250 U/g (St. Louis, MO, USA). All materials were food-grade ingredients.

### 2.1. Enzymatic Hydrolysis of Corn Starch

To avoid possible traces of the acids used for starch hydrolysis, it was decided to perform starch hydrolysis with enzymes. The starch used was from corn, with a prior selection among all sources available on the Romanian market based on the analysis of different properties. Based on the method described by Kong et al. with some minor modifications, enzymatic hydrolysis of corn starch was performed with a-amylase from *Bacillus amyloliquefaciens* [27]. Corn starch was added to water and homogenized with a vortex for 5 min. The mixture was heated in a water bath at 60 °C for 15 min with stirring. Then, the enzyme was added and the mixture was incubated at a controlled temperature for 1 h under homogenization at 160 rpm. Thus, the ratio of solid loading varied according to the volume of liquid used and ranged from 0.25 g/mL to 0.5 g/mL. After the time elapsed, the mixture was suddenly cooled in an ice water bath to inactivate the enzyme. The corn-starch-to-water ratio, the enzyme-to-substrate ratio, and the incubation temperature were adjusted to three different levels (Table 1).

The resulting solution was centrifuged at least 4 times for 15 min at 9000 rpm. A sample was taken from the first supernatant, and the reducing sugars were determined to calculate the DH, and the remaining supernatant was removed. To the precipitate, 100 mL of distilled water was added and washed several times until a pH of 6–7 was reached. Finally, the precipitate was recovered and dried at 40 °C (hydrolyzed corn starch).

### 2.2. Optimization of Mechanical Properties Using RSM with BBD Experimental Design

The formulation (1:2–1:4 corn-starch-to-distilled-water ratio, 250–500 U/g enzyme-to-substrate ratio) and incubation temperature of 40–50 °C to give the optimal DH for hydrolyzed corn starch to obtain improved mechanical properties of its film-forming solutions were determined using RSM techniques. RSM was applied to investigate the effects of the independent variables (corn-starch-to-distilled-water ratio (A), enzyme-to-substrate ratio (B), and incubation temperature (C)) on the responses: DH (Y_1_) for the hydrolyzed corn starch and tensile strain at break (Y_2_), tensile stress at break (Y_3_) and Young’s modulus (Y_4_) for the hydrolyzed corn starch films.

RSM with BBD of three independent variables and their combinations was used in this experiment to determine the formulation for the optimal hydrolyzed corn starch and hydrolyzed corn starch films in terms of DH (Y_1_), tensile strain at break (Y_2_), tensile stress at break (Y_3_) and Young’s modulus (Y_4_). All independent variables, including corn-starch-to-water ratio, enzyme-to-substrate ratio, and incubation temperature, were performed at three levels in terms of BBD (Table 1) for each coded value (−1, 0, and 1). The BBD was performed to investigate the effect of each independent variable and the interaction between the independent variables on the response. A second-order polynomial equation was used to indicate the responses as a function of the independent variables (Equation (1)).
Y_i_ = a_0_ + a_1_A + a_2_B + a_3_C + a_11_A^2^ + a_22_B^2^ + a_33_C^2^ + a_12_AB + a_13_AC + a_23_BC(1)
where Y_i_ is the response function (i = 1, 2, and 3), a_0_ is a constant, and the coefficients in Equation (1) are linear (a_1_, a_2_, and a_3_), quadratic (a_11_, a_22_, and a_33_), and interaction (a_12_, a_13_, and a_23_) coefficients, respectively.

RSM with three coded levels (−1, 0, and 1) was employed to examine the effects of corn-starch-to-distilled-water ratio (A: 1:2–1:4 g/mL), enzyme-to-substrate ratio (B: 250–500 U/g), and temperature (C: 40–50 °C) on the DH (Y_1_), tensile strain at break (Y_2_), tensile stress at break (Y_3_) and Young’s modulus (Y_4_) for the hydrolyzed corn starch films. The actual values and coded factor values of the three independent variables for optimizing the formulation of hydrolyzed corn starch films using RSM are shown in Table 2.

The RSM with BBD with 3 coded levels, 3 independent variables, and 15 experimental runs with three midpoints (Table 2) was used to optimize enzymatic process conditions, mainly corn-starch-to-water ratio (A), enzyme-to-substrate ratio (B), and incubation temperature (C) of hydrolyzed corn starch films using Design-Expert^®^ software (version 13.0, Stat Soft Inc., Tulsa, OK, USA) (for now, the corn-starch-to-water ratios are expressed as 2, 3, and 4 in this article instead of 1:2, 1:3, and 1:4, as this simplifies the process of graphical and statistical analysis).

### 2.3. Determination of the DH

The indicator of starch hydrolysis is the DH value, calculated according to Equation (2) [28]:(2)$$\mathrm{DH}\;\left(\%\right)=\frac{\mathrm{amount}\;\mathrm{of}\;\mathrm{reducing}\;\mathrm{sugars}\;\mathrm{released}\;\mathrm{during}\;\mathrm{enzymatic}\;\mathrm{hydrolysis}}{\mathrm{initial}\;\mathrm{amount}\;\mathrm{of}\;\mathrm{starch}\;\mathrm{substrate}}\;\times \;100$$

### 2.4. Determination of the Reducing Sugars

The determination of reducing sugar content in hydrolyzed corn starch samples was performed using a modified version of the qualitative Benedict technique [29] based on the protocol introduced by Hernández-López et al. [30]. The method is described as a simple, rapid, and reliable method for quantifying reducing sugars in various samples, including foods. Corn starch samples were mixed with Benedict reagent, and the resulting mixture underwent a redox reaction with reducing sugars after the solution was heated at 70 °C for a few minutes, producing an orange-brown color and a precipitate. After cooling the samples, the absorbance of the color change was quantified using a Shimazu spectrophotometer at λ = 575 nm. The content of reducing sugars was expressed as milligrams of glucose equivalents per gram of sample. The modified protocol proved to be simple and effective in determining the content of reducing sugars in our hydrolyzed corn starch samples.

### 2.5. Preparation of Hydrolyzed Corn Starch Films

The casting technique was used to prepare films of hydrolyzed corn starch powder, glycerol, and glacial acetic acid according to the methodology of Beer-Lech et al. [31]. First, 20 ± 5 g of hydrolyzed corn starch powder was mixed with 150 ± 5 mL of distilled water at 50 ± 5 °C to prepare the film-forming solution. Glycerol was added as a plasticizer at a ratio of 1:2 *v*/*w* concerning the hydrolyzed corn starch powder, along with 0.5 M acetic acid at a ratio of 1:2 *v*/*w* with respect to the hydrolyzed corn starch powder. The resulting mixture was heated to 70 ± 5 °C with constant stirring using a magnetic stirrer at 300 rpm until complete gelation occurred. The solution was then cooled to 50 °C and poured into 20 × 20 cm^2^ glass plates to ensure uniform thickness of the film. The films were allowed to dry at room temperature of 25 ± 5 °C for about 72 h. They were then removed from the glass plates and stored in a desiccator for characterization.

### 2.6. Mechanical Properties

The mechanical properties of hydrolyzed corn starch films were evaluated by the method of Panaitescu et al. [32], with minor changes, at room temperature using an Instron 3382 (Instron, University Ave, Norwood, MA, USA) with a 10 kN load cell according to ISO 527-3:1995. The specimens were fixed with a dumbbell-shaped clamp with a length of 50 mm, a width of 8 mm at the ends, and a width of 4 mm at the narrow part. Tensile strain at break, tensile stress at break, and Young’s modulus were determined at a test speed of 2 mm/min. Each specimen was evaluated using five specimens, and the average values for each specimen were then reported.

### 2.7. Water Absorption Index and Transparency Measurements

#### 2.7.1. Water Absorption Index

The determination of the water absorption index was performed according to the method described by Diop et al. with minor modifications [33]. A total of 2 g of corn starch and hydrolyzed corn starch were added to a 50 mL Falcon tube, over which distilled water was added to a total volume of 40 mL. The suspension was shaken at low speed for 2 min to avoid the fragmentation of the starch granules. The samples were then centrifuged at 3000 rpm for 10 min. The supernatant was removed, and the wet pellets were weighed. The water absorption index was calculated according to Equation (3):(3)$$\mathrm{Water}\;\mathrm{absorption}\;\mathrm{index}\;=\frac{\mathrm{Weight}\;\mathrm{of}\;\mathrm{hydrated}\;\mathrm{precipitate}\;-\;\mathrm{Weight}\;\mathrm{of}\;\mathrm{sample}}{\mathrm{Weight}\;\mathrm{of}\;\mathrm{sample}}$$

#### 2.7.2. Transparency Measurements

The transparency of the starch films was evaluated by measuring the optical transmittance and absorbance using a UV–Visible spectrophotometer (Shimadzu Europa GmbH, Duisburg, Germany). To perform the analysis, 10 × 10 mm^2^ of the sample was placed in a quartz cuvette and subjected to measurements in the range of 200–400 nm. The results were expressed as a percentage of transmittance per millimeter [34].

### 2.8. FTIR Analysis

To investigate the changes in molecular structure due to hydrolysis, FTIR spectra of corn starch films and hydrolyzed corn starch films were recorded using a Bruker Tensor 27 (Bruker, Germany) spectrometer with a ZnSe ATR accessory. The analyzed samples were coated directly onto the ATR ZnSe crystal, which was cleaned with ethanol to remove residues before each measurement. Spectra were recorded in the wavelength range 650–4000 cm^−1^ with 32 scans per sample, a resolution of 4 cm^−1^, and a velocity of 0.32 cm/s.

### 2.9. Water Contact Angle

The CAM 200 contact angle tensiometer (KSV Instruments Ltd., Espoo, Finland) was used to measure static contact angles at room temperature. The instrument was equipped with a high-resolution camera (Basler A602f) and an auto-dispenser. To measure contact angles, 5 liquid drops with a drop volume of 6 µL were placed on different areas of the film surface. The resulting contact angles were calculated from all photos of the water droplet [35].

### 2.10. Differential Scanning Calorimetry (DSC)

DSC analyses were performed using the TA Q2000 instrument (DSC, Q2000, TA instruments, New Castle, DE, USA) with hermetically sealed aluminum trays while samples were under a helium flow of 25 mL/min. The thermal behavior of the samples was monitored by cooling them from room temperature to −75 °C, followed by heating them to 250 °C at a rate of 10 °C/min. An amount of 16–17 mg was used for each sample, and all tests were repeated at least three times [36].

### 2.11. AFM Characterization

AFM characterization (AFM, NT-MDT Spectrum Instruments, Moscow, Russia) with a Pro m NT-MDT system in noncontact mode was used to study morphology and measure surface roughness (NC-AFM). Random zones were scanned with the AFM. Nova software was used to process the 2D AFM images. The following statistical characteristics associated with the surface roughness of the sample were estimated according to ASME B46.1 (ASME, 1995): mean roughness (Sa: average of the absolute value of the height deviations from a mean surface); squared roughness (Sq: square root of the height deviations obtained from the mean data plane). To obtain these parameters, three repetitions were considered [37].

### 2.12. Dynamic Mechanical Analysis (DMA)

The dynamic mechanical analysis was carried out using a DMA Q800 instrument (TA Instruments, New Castle, DE, USA), assisted by a liquid nitrogen pump with a heating rate of 3 °C/min^−1^ in DMA multiple Frequency strain with film tension clamp. Duplicate samples (12.75 × 6.3 × 0.3 mm^3^) were scanned over a temperature range from −80 °C to 120 °C at a frequency of 1 Hz and 15 µm amplitude [35].

### 2.13. Statistical Analysis

To examine the relationship between a response and two independent variables, three-dimensional surface plots were created using Design-Expert^®^ software (version 13.0, Stat Soft Inc., Tulsa, OK, USA) in a stepwise procedure. A model term was considered significant if its *p* value was less than 0.05. Data were analyzed using Minitab 20 statistics software (Minitab LLC, State College, PA, USA), and the Tukey comparison test was used to detect significant differences (*p* > 0.05). Results were expressed as mean ± standard deviation of at least three measurements (n = 3).

## 3. Results and Discussions

### 3.1. Optimization of Enzymatic Hydrolysis Parameters Using RSM and BBD

The values for experimental, predicted, and residual for each response of the hydrolyzed corn starch and hydrolyzed corn starch films are listed in Table 3.

The experimental (actual) and anticipated values of the responses showed remarkable agreement. The output of the statistical results confirmed the significance of the proposed model (Table 3). The coefficient of determination confirmed the accuracy of the developed model. Table 4 shows the coefficient of multiple characterizations (R^2^) and the adjusted coefficient of multiple characterizations (adjusted R^2^) of DH, the tensile strain at break, the tensile stress at break, and Young’s modulus, which indicate whether a regression model is appropriate. ANOVA was used to determine the significance of the coefficient of the quadratic polynomial equations. A large F value and a small *p* value indicate a significant effect of each term [38].

DH was found to be influenced primarily by the linear influence of the corn-starch-to-water ratio and by the interaction effect between the corn-starch-to-water ratio and the enzyme-to-substrate ratio, in descending order (*p* ≤ 0.01). Thereafter, the quadratic effect of the corn-starch-to-water ratio and the interaction effect between the enzyme-to-substrate ratio and the incubation temperature had a significant effect on DH (*p* ≤ 0.05). In contrast, the DH was not influenced by the interaction effect between the corn-starch-to-water ratio and the incubation temperature, nor by the two linear effects and the two quadratic effects of enzyme-to-substrate ratio and incubation temperature. Therefore, the final models included only the significant terms. The model was considered to have a good fit because the test for lack of fit (*p* > 0.05) was not significant, indicating the high efficiency of the quadratic model in fitting the data under experimental conditions. In addition, high R^2^ values indicated a good fit of the model, whereas the fitted R^2^ values indicated a good agreement between the predicted and experimental values for DH (Table 4).

Using Table 4, it was found that the variables with the largest to smallest effects on the tensile strain at break were the linear effect of corn starch to water ratio, the linear effect of incubation temperature, and the quadratic effects of all independent variables (*p* ≤ 0.01). Conversely, the linear effect of the enzyme-to-substrate ratio and the interaction effects of all independent variables were found to have no effect on the tensile strain at break, and were therefore excluded from the final models. The model was considered to fit the data well because the test for lack of fit (*p* > 0.05) was not significant for the tensile strain at break (Table 4). In addition, the high R^2^ values indicated that the quadratic model fit the data very well under the experimental conditions, while the fitted R^2^ values indicated a good agreement between the predicted and experimental values of the model for the tensile strain at break.

Table 4 shows that, for the tensile stress at break, the variables with the largest to smallest effects on the value were the linear effect of the incubation temperature, the linear effect of the enzyme-to-substrate ratio, and the interaction effect between the corn-starch-to-water ratio and the enzyme-to-substrate ratio (*p* ≤ 0.01). The quadratic effect of the enzyme-to-substrate ratio also affected the tensile stress at break (*p* ≤ 0.05). However, the linear effect of the corn-starch-to-water ratio, the residual interaction effects of the three independent variables, the quadratic effect of the corn-starch-to-water ratio, and the quadratic effect of the incubation temperature showed an insignificant effect (*p* > 0.05) and were therefore excluded from the final models. The model was considered to have a good fit because the test for lack of fit (*p* > 0.05) for the tensile stress at break indicated a good fit (Table 4). High R^2^ values indicated that all quadratic models fitted the data very well under the experimental conditions, while the fitted R^2^ values indicated a good agreement between the predicted and experimental values of the model for the tensile stress at break (Table 4).

According to Table 4, the variables with the largest to smallest effects on the Young’s modulus were the linear effect of incubation temperature, followed by the interaction effect between the corn-starch-to-water ratio and the enzyme-to-substrate ratio (*p* ≤ 0.01). In addition, the linear effect of the corn-starch-to-water ratio, the linear effect of the enzyme-to-substrate ratio, and the interaction effect between the corn-starch-to-water ratio and the incubation temperature affected the Young’s modulus (*p* ≤ 0.05). The test for lack of fit (*p* > 0.05) for the Young’s modulus was not significant, indicating that the model was a good fit (Table 4). The high R^2^ values indicated that all quadratic models fitted the data very efficiently under the experimental conditions, whereas the fitted R^2^ values indicated a good fit between the predicted and experimental values of the model for the Young’s modulus (Table 4). Table 5 shows the second-order polynomial equations for the models and the regression coefficients of the polynomial response surface models for the responses.

After analyzing the experimental data obtained from the independent variables, it was found that the response surface plots for the DH, tensile strain at break, tensile stress at break, and Young’s modulus of both the hydrolyzed corn starch and hydrolyzed corn starch films could be plotted using a quadratic polynomial equation for prediction purposes.

A response surface plot was created using Design-Expert^®^ software to show the effects of the corn-starch-to-water ratio, enzyme-to-substrate ratio, and incubation temperature on the response variables. These plots were created by changing two independent variables within the experimental ranges while maintaining the mean value of the third variable.

Figure 1, Figure 2, Figure 3 and Figure 4 show the response surfaces plots of the quadratic and 2FI polynomial models representing the influence of the independent variables on the DH, tensile strain at break, tensile stress at break, and Young’s modulus of hydrolyzed corn starch and its films.

Figure 1 shows the surface plots of the three-dimensional response for the DH of hydrolyzed corn starch.

The influence of the corn-starch-to-water ratio and the enzyme-to-substrate ratio on the DH reaction of hydrolyzed corn starch was investigated using a 3D diagram (Figure 1a). The graph showed that DH increased with an increasing enzyme-to-substrate ratio, while it decreased with a decreasing corn-starch-to-water ratio. This result is consistent with previous studies reporting that the enzyme-to-substrate ratio is a key factor affecting the DH of hydrolyzed starch [39,40]. Moreover, the effect of the corn-starch-to-water ratio on the DH response was in agreement with the results of Cornejo et al. [41], who reported that a higher water content in the reaction mixture can lead to a higher DH. These results suggest that optimizing the enzyme-to-substrate ratio and the corn-starch-to-water ratio can significantly increase the DH of hydrolyzed corn starch.

The effect of the corn-starch-to-water ratio and the incubation temperature on the DH of hydrolyzed corn starch was investigated, and the results are shown in the 3D response surface plot in Figure 1b. The reaction increased with an increase in the corn-starch-to-water ratio and the incubation temperature. On the other hand, a decrease in the corn-starch-to-water ratio resulted in a decrease in DH. These results are consistent with previous studies reporting that increasing the substrate concentration can increase the DH [42]. In addition, the incubation temperature is a crucial factor affecting the enzymatic hydrolysis process [43]. In our study, the quadratic polynomial model was used to generate the response surface plots that allowed us to determine the optimal conditions for maximizing DH. This approach has been widely used in previous studies to optimize the enzymatic hydrolysis conditions for different substrates [44].

The effect of the enzyme-to-substrate ratio and incubation temperature on DH is shown in the 3D response surface plot shown in Figure 1c. During the enzyme-catalyzed hydrolysis of starch, the α-1,4-glycosidic bonds in the starch molecules are cleaved, and the reaction rate is affected by the enzyme concentration and temperature [45,46]. The results show that increasing both the enzyme-to-substrate ratio and the incubation temperature can increase the DH, which is consistent with previous studies on the enzymatic hydrolysis of starch [47,48]. On the other hand, decreasing the enzyme-to-substrate ratio can decrease the DH, which is consistent with the finding of Mora S. [49], who reported that the DH of corn starch decreased when the enzyme-to-substrate ratio was below the optimal range. The 3D response surface diagrams for tensile strain at the break of the hydrolyzed corn starch film are shown in Figure 2.

Figure 2a shows the effects of the corn-starch-to-water ratio and the enzyme-to-substrate ratio during enzymatic hydrolysis on the tensile strain at the break of the hydrolyzed corn starch film. The diagram shows that the response increases with the corn-starch-to-water ratio, and reaches a maximum at an intermediate value of the corn-starch-to-water ratio and decreases with an increase in the enzyme-to-substrate ratio during enzymatic hydrolysis. Moreover, the graph shows that the response is highest when both variables are at intermediate values. This shows that it is important for the tensile strain at the break of the hydrolyzed corn starch film to have an optimum range for the variables during enzymatic hydrolysis to obtain the maximum response [50].

The effect of the corn-starch-to-water ratio and the incubation temperature on the tensile strain at the break of films of hydrolyzed corn starch is shown in the 3D reaction plot in Figure 2b. The plot shows that an increase in both variables at intermediate values during enzymatic hydrolysis leads to an increase in the tensile strain at the break of the film. Furthermore, an increase in the corn-starch-to-water ratio during enzymatic hydrolysis also leads to an increase in the tensile strain at break. This can be attributed to the fact that a higher ratio of corn starch to water results in better accessibility for the enzyme to be more active, resulting in increased mechanical strength and better film-forming ability of the corn starch after hydrolysis [51]. However, increasing the incubation temperature leads to a decrease in the tensile strain at break. This could be due to the thermal degradation of the corn starch molecules under the action of the enzyme and the subsequent weakening of the starch structure [52]. Overall, the 3D reaction diagram shows the complex relationship between the corn-starch-to-water ratio, the incubation temperature during enzymatic hydrolysis, and the tensile strain at the break of films of hydrolyzed corn starch. These results demonstrate the importance of optimizing the enzymatic hydrolysis process to achieve the desired film properties for various applications.

The effect of the enzyme-to-substrate ratio and the incubation temperature on the tensile strain at the break of the film of hydrolyzed corn starch is shown in Figure 2c. The results show that the response increases with an increase in incubation temperature, while it decreases with an increase in the enzyme-to-substrate ratio during enzymatic hydrolysis. This result is consistent with previous studies, where an increase in temperature during enzymatic hydrolysis could lead to an improvement in the tensile properties of corn starch films. This is due to the increased mobility of the starch chains and the improved molecular interactions within the starch structure, resulting in improved film formation properties [53].

The 3D response surface diagrams for tensile stress at the break of the hydrolyzed corn starch film are shown in Figure 3.

From Figure 3a, it can be seen that the tensile stress at the break of the film of hydrolyzed corn starch increases as both the corn-starch-to-water ratio and the enzyme-to-substrate ratio increase. This is consistent with previous studies that have shown that enzyme treatment can improve the tensile properties of starch films [41]. The increase in tensile stress at break can be attributed to the change in starch structure and the formation of stronger intermolecular bonds between the starch molecules [54]. In addition, the increase in the corn-starch-to-water ratio during enzymatic hydrolysis can result in an improved molecular structure, which can help to improve tensile strength properties [55]. These results suggest that adjusting the corn-starch-to-water ratio and the enzyme-to-substrate ratio during the process may be an effective strategy for optimizing the tensile strength of hydrolyzed corn starch films.

From Figure 3b, it can be seen that the tensile stress at the break of the hydrolyzed corn starch film is affected by both the corn-starch-to-water ratio and the incubation temperature during enzymatic hydrolysis. The response increases as the incubation temperature increases and decreases as the corn-starch-to-water ratio decreases. Therefore, the present study suggests that the optimization of the incubation temperature and the corn-starch-to-water ratio during the enzymatic hydrolysis of corn starch can lead to improved mechanical properties of hydrolyzed corn starch films.

The effect of the enzyme-to-substrate ratio and the incubation temperature on tensile stress at the break of hydrolyzed corn starch film is shown in Figure 3c. The results show that the response increases with an increase in incubation temperature, while it decreases with an increase in the enzyme-to-substrate ratio during enzymatic hydrolysis. This result is consistent with previous studies in which an increase in temperature during enzymatic hydrolysis could lead to an improvement in the tensile strength of corn starch films. This is due to the increased mobility of the starch chains and the improved molecular interactions within the starch structure, resulting in improved film formation properties [56]. On the other hand, an increase in the enzyme-to-substrate ratio during enzymatic hydrolysis can lead to a decrease in the tensile strength of films of hydrolyzed corn starch, since excessive enzyme concentrations can lead to the degradation of starch molecules and a weakening of the properties of the starch film [57]. This study suggests that optimization of the incubation temperature and the enzyme-to-substrate ratio during the enzymatic hydrolysis of corn starch can lead to the production of corn starch films with desirable elongation properties.

The 3D response surface diagrams for tensile stress at the break of the film of hydrolyzed corn starch are shown in Figure 4.

The three-dimensional response diagrams for the Young’s modulus of the hydrolyzed corn starch film as a function of the corn-starch-to-water ratio and enzyme-to-substrate ratio are shown in Figure 4a. The 3D plots show that increasing the corn-starch-to-water ratio can increase the Young’s modulus. This is consistent with previous studies that reported an increase in mechanical properties with increasing starch concentration [58]. On the other hand, the reaction decreases when the ratio of enzyme to substrate increases. This can be attributed to the fact that enzymes can degrade starch and reduce its mechanical strength. The observed trend is consistent with previous studies reporting the effects of enzyme concentration on starch degradation and mechanical properties of starch-based films [59].

The 3D response diagram for the Young’s modulus of the hydrolyzed corn starch film as a function of the corn-starch-to-water ratio and incubation temperature during enzymatic hydrolysis is shown in Figure 4b. The plot shows that the Young’s modulus of the film decreases as the corn-starch-to-water ratio increases during enzymatic hydrolysis, while it increases as the incubation temperature increases. The effect of the corn-starch-to-water ratio and the incubation temperature on the mechanical properties of starch-based films has not yet been studied, but it can be reported that increasing the corn-starch-to-water ratio during enzymatic hydrolysis leads to the formation of more ordered and compact structures, resulting in a stiffer film with a higher Young’s modulus [60]. On the other hand, an increase in the incubation temperature during enzymatic hydrolysis can cause the rearrangement of starch chains, leading to a decrease in intermolecular forces [61] and thus the Young’s modulus of the hydrolyzed corn starch film. Therefore, the observed behavior in the 3D diagram can be attributed to the changes in the internal structure of corn starch caused by the variation of the parameters of the enzymatic hydrolysis process. Overall, the presented 3D reaction diagram provides valuable insights into the relationship between the enzymatic hydrolysis process, the corn starch structure, and the properties of the hydrolyzed corn starch film, and can be used to optimize the film properties.

The effect of the enzyme-to-substrate ratio and the incubation temperature on the Young’s modulus of the hydrolyzed corn starch film is shown in the 3D graph in Figure 4c. The graphs show that an increase in both variables during enzymatic hydrolysis leads to an increase in the Young’s modulus of the film. This result shows that the use of enzymes can increase the strength and elasticity of films made from hydrolyzed corn starch. It was also reported that the incubation temperature can affect the mechanical properties of starch films by influencing the formation of the amylose–lipid complex during enzymatic hydrolysis [62]. Interestingly, our results indicate that the Young’s modulus of the hydrolyzed corn starch film further increases as both variables increase during enzymatic hydrolysis, suggesting that there may be a synergistic effect of the enzyme-to-substrate ratio and the incubation temperature on the properties of the hydrolyzed corn starch film. This result is consistent with previous studies that have shown that the use of enzymes in combination with heat treatment can improve the film-forming properties of corn starch [63]. This finding highlights the potential of enzymatic hydrolysis treatments of starch to improve the mechanical properties of starch-based materials, which has important implications for the development of sustainable packaging materials.

#### 3.1.1. Optimal Independent Variables

The desirability function was then used in conjunction with Design-Expert^®^ software to perform the numerical optimization. The corn-starch-to-water ratio, enzyme to substrate ratio, and incubation temperature had to be within the concentration range at which the films were prepared to maximize mechanical properties. These were the goals chosen for the optimization of hydrolyzed corn starch films. Ten different solutions were found, each with a different level of independent variables. The best option for optimizing the mechanical properties of hydrolyzed corn starch films was selected because it had the highest desirable value (0.857). Table 6 shows the optimum conditions and experimental and predicted values of responses at optimized conditions.

The optimized conditions for hydrolyzed corn starch film were a 1:2.8 corn-starch-to-water ratio, a 357 U/g enzyme-to-substrate ratio, and a 48 °C incubation temperature.

#### 3.1.2. Verification of the RSM Model Based on DH and Mechanical Properties

Optimal enzymatic hydrolysis conditions were used to test the suitability of the model for predicting optimal mechanical properties. Experiments under ideal enzymatic hydrolysis conditions were used to validate the optimum DH of hydrolyzed corn starch to obtain the improved mechanical properties. The mechanical properties of hydrolyzed corn starch were determined under ideal DH conditions when it was produced in the form of a packaging film. In contrast, the experimental results were obtained under optimal enzymatic hydrolysis conditions. The optimal predicted values of mechanical properties and the optimal experimental values were in good agreement (Table 6).

### 3.2. Water Absorption Index and Transparency Measurements

#### 3.2.1. Water Absorption Index

The water solubility index of starch is a measure of its ability to dissolve in water and is influenced by factors such as the structure of the starch granules, the ratio of amylose to amylopectin, and the degree of branching. Native corn starch has a compact and ordered granular structure that limits the amount of water that can enter the grains and interact with the starch molecules [64]. The hydrolysis of starch by enzymes breaks down the structure of grains and reduces their size, creating a larger surface area for water to interact with starch molecules [65].

As a result, hydrolyzed corn starch film (2.32 ± 0.112%) has a higher water absorption index value than native corn starch film (0.81 ± 0.352%) because its modified granular structure makes it more water-soluble.

#### 3.2.2. Transparency Measurements

The transparency of a starch film is determined by the degree of order and arrangement of starch molecules within the film matrix. Native corn starch has a highly organized and compact granular structure that limits the mobility of starch molecules within the film [58]. Enzymatically hydrolyzed corn starch, on the other hand, has a more open and disorganized granular structure due to the action of the enzymes, resulting in a more flexible and mobile film matrix [66]. This increased mobility allows light to pass through the film with less scattering, resulting in a more transparent film. In addition, the hydrolysis process also reduces the molecular weight of the starch, which also contributes to increased transparency due to the reduced intermolecular interactions and the increased mobility of the starch molecules [67].

Therefore, films made from enzymatically hydrolyzed corn starch tend to be more transparent (78.5 ± 0.121% transmittance/mm) than films made from native corn starch (65.6 ± 0.230% transmittance/mm) due to their altered granular structure and lower molecular weight.

### 3.3. FTIR Analysis

FTIR is a technique for identifying the functional groups present in a sample and provides information about the chemical bonding of the sample (Figure 5) [68]. Enzymatically hydrolyzed corn starch films with glycerol as a plasticizer have a more compact and firmer structure compared to native corn starch films due to the presence of more hydrogen bonds between the starch molecules. The hydrolysis of corn starch releases more hydroxyl groups on the starch molecules, which can interact with the hydroxyl groups of glycerol via hydrogen bonds [69].

The increased number of hydrogen bonds between the starch molecules and glycerol resulted in a more densely packed and ordered film structure, as shown by FTIR analysis. Native corn starch films, on the other hand, have fewer hydroxyl groups on the surface of the starch molecules, which resulted in fewer hydrogen bonds and a more disordered film structure. In the case of the hydrolyzed corn starch film, the presence of multiple peaks at 1456 cm^−1^, ~1100 cm^−1^, and ~900 cm^−1^, not found in other samples, could be due to the specific chemical composition of the hydrolyzed corn starch film. Hydrolyzed corn starch films likely contain unique functional groups due to the hydrolysis process that breaks the larger polysaccharide chains into smaller ones. This process may lead to the formation of new chemical bonds such as carboxyl, hydroxyl, and aldehyde groups, which are responsible for the appearance of the peaks at 1456 cm^−1^, ~1100 cm^−1^, and ~900 cm^−1^ in the FTIR spectrum. The peak at 1456 cm^−1^ can be attributed to the asymmetric bending of the CH_2_ groups in the acetyl groups, which often occurs in starch. The peak at ~1100 cm^−1^ can be attributed to the C–O–C stretching vibrations typical of the glycosidic bonds connecting the glucose units in the starch molecule. Finally, the peak at ~900 cm^−1^ can be attributed to C–H bending vibrations indicating the presence of alkyl groups. Therefore, enzymatically hydrolyzed corn starch films with glycerol as a plasticizer have a more compact and solid structure in terms of molecular bonds compared to native corn starch films, as shown by FTIR analysis.

### 3.4. Water Contact Angle

The determination of the contact angle can provide information about the hydrophilic and hydrophobic properties of the surface of films [70]. Contact angles greater than 90° indicate a hydrophobic character of the surface, while angles less than 90° indicate a hydrophilic character [71]. In this study, enzymatically hydrolyzed corn starch films showed an increased contact angle compared to native corn starch films. This can be attributed to the change in starch structure caused by enzymatic hydrolysis, which resulted in a change in the hydrophilic character of the films.

Figure 6 shows that the contact angle was higher for hydrolyzed corn starch films (79.21 ± 0.171°) compared to the native corn starch (63.97 ± 0.232°). The hydrolysis process increases the number of hydroxyl groups available for multiple bonds, which facilitates gelation and the formation of a 3D matrix of the film due to the formation of hydrogen bonds between the available hydroxyl groups and the plasticizer. This can lead to a more cohesive 3D network and the use of OH for hydrogen bond formation, increasing the hydrophobicity of the polymer matrix and the contact angles of the films.

Wu et al. also observed an increased contact angle when glycerol was used as a crosslinking agent in polysaccharide films [72]. The differences in contact angle between native corn starch and hydrolyzed corn starch films can be explained by differences in the hydrolysis process, with the hydrolytic process applied to the polymer leading to more aggressive degradation of the amylose and amylopectin structures. This results in a less homogeneous surface area and larger amounts of hydroxyl groups available for reaction with water, leading to a larger contact angle [73].

The determination of the contact angle can indicate the hydrophobic and hydrophilic properties of the surface of films, and the strategies applied to the polymer can change the hydrophilic character of the films, increasing the water contact angle.

### 3.5. DSC Thermal Analysis

Figure 7 shows the DSC heating scans of native corn starch and hydrolyzed corn starch films.

Two endothermic events were observed in the DSC thermograms of the two films. The first endothermic event could be observed between 169.2 °C and 199.4 °C for hydrolyzed corn starch films and between 194 °C and 224.8 °C for native corn starch films. This first endothermic event could be related to starch melting, as noted by other authors [74,75]. The broad endothermic event observed in both corn-starch-based films was likely a complex event resulting from the overlap of several events, including starch melting and water release, which could mask the glass transition. The significant difference in the temperature of the first endothermic event between the two films indicated that the native corn starch film had a higher melting point than the hydrolyzed corn starch film. The loss of water from the amorphous phase of starch could enhance hydrogen bonding, leading to an increase in the glass transition temperature (Tg), from −67.6 °C to −49.3 °C for native corn starch film, and also prolonging the glass transition. This could be due to differences in the crystalline structure and molecular weight of the starch in the two films. The shift of the first endothermic peak to a higher temperature in the native corn starch film compared to the hydrolyzed corn starch film could also be related to the limited segment mobility due to crosslinking with glycerol. Crosslinking is known to increase Tg in synthetic polymers, and some starch materials have also been reported [76]. During the enzymatic hydrolysis of corn starch, long-chain starch molecules are broken down into shorter-chain molecules by enzymes. This process can lead to a reduction in the molecular weight and degree of crystallinity of the starch, which can ultimately affect its thermal properties.

In the case of DSC scans of hydrolyzed corn starch films, the smaller first endothermic event at 169.2 °C and 199.4 °C compared to the native corn starch film at 194 °C and 224.8 °C could be due to the lower molecular weight and degree of crystallinity resulting from enzymatic hydrolysis. The lower molecular weight may result in fewer crystalline regions in the starch, resulting in a reduction in the heat required for melting.

Enzymatic hydrolysis may also result in the formation of shorter chain molecules with greater segmental mobility, lowering the Tg and reducing the temperature required for water release during the glass transition from −65.3 °C to −44.5 °C. This could also contribute to the lower endothermic event observed in the hydrolyzed corn starch film [77].

### 3.6. AFM Characterization

Surface roughness plays a critical role in determining the physical, chemical, and biological properties of films, making its quantitative analysis essential for the development and optimization of film-based products. AFM is a powerful tool for characterizing surface roughness, providing detailed information on roughness parameters such as Sq and Sa. In this context, AFM characterization of a hydrolyzed corn starch film was performed to investigate the effect of plasticizer concentration on the surface roughness. This would provide valuable insights into the relationship between the choice of plasticizer and the surface roughness of corn-starch-based films, which could be used to develop films with tailored properties. Table 7 presents the roughness parameters (Sq, Sa) of hydrolyzed corn-starch-based films that were plasticized with glycerol, which were determined by an analysis of AFM images.

Figure 8 shows two-dimensional images of the films, which provide visual information about the surface topography of the samples.

The AFM characterization of hydrolyzed corn starch film revealed that the roughness parameters (Sq, Sa) were in the range of those observed in a study of corn-starch-based films plasticized with different glycerin ratios. The hydrolyzed corn starch film exhibited Sq values between 60 and 90 nm, indicating an intermediate surface roughness. This result was consistent with a previous study by Wang B et al. [78], in which it was suggested that glycerol content may be the main factor contributing to increased roughness in starch-based films. In addition, it was found that phase separation between the starch and the plasticizer can increase surface roughness, which emphasizes the importance of selecting an appropriate plasticizer to optimize the properties of corn-starch-based films.

### 3.7. Dynamic Mechanical Analysis (DMA)

Figure 9 and Table 8 shows the storage modulus (E′) and damping factor (tan δ) spectra of the native corn starch film and hydrolyzed corn starch film over a temperature range from −50 to 150 °C.

Comparing the data of the two samples, we found that both had a similar composition (corn starch film with glycerol as plasticizer) but slightly different DMA results. The initial, mean, and final temperatures for the hydrolyzed corn starch film sample were all slightly lower than those for the native corn starch film sample (−63.66 °C vs. −62.11 °C, −53.3 °C vs. −49.57 °C, and −44.35 °C vs. −37.72 °C, respectively). This indicated that the storage modulus of the hydrolyzed corn starch film sample changed over a wider temperature range than that of the native corn starch film sample. The temperature at which the loss modulus was highest was also slightly lower for the native corn starch film sample (−50.61 °C vs. −61.58 °C), although the actual value of the loss modulus was also lower for the native corn starch film sample (164.9 MPa vs. 304.6 MPa). This suggests that the native corn starch film sample was less capable of dissipating energy and deforming under load than the hydrolyzed corn starch film. The temperature at which the Tan Delta was highest was also slightly lower for the native corn starch film (−43.35 °C vs. −51.09 °C), although the actual value of the Tan Delta was also lower for the hydrolyzed corn starch film (0.08889 vs. 0.1372). This indicates that the native corn starch film sample had lower energy dissipation properties than the hydrolyzed corn starch film. The hydrolyzed corn starch film appeared to have better mechanical properties than the native corn starch film sample, with a greater change in storage modulus over a wider temperature range and higher values for loss modulus and Tan Delta. Additionally, the storage modulus of the native corn starch film was higher than that of the hydrolyzed corn starch film in the temperature range of interest, indicating that the native corn starch film had a stiffer structure. The damping factor of both films decreased with increasing temperature, indicating that the ability of the films to dissipate energy decreases with increasing temperature.

The difference in storage modulus between the two films can be attributed to differences in the molecular weight and crystalline structure of the starch in the films. The hydrolysis process can split the starch molecules into smaller fragments, resulting in a decrease in molecular weight and an amorphous structure. This can result in a lower storage modulus and higher flexibility of the hydrolyzed corn starch film compared to the native corn starch film.

## 4. Conclusions

In conclusion, BBD and RSM were used in the study to optimize the enzymatic hydrolysis process for corn starch films. The results of the study showed that the ratio of corn starch to water, the ratio of enzyme to substrate, and the incubation temperature significantly affected the mechanical properties of the hydrolyzed corn starch films. The optimum conditions were found to be a corn-starch-to-water ratio of 1:2.8, an enzyme-to-substrate ratio of 357 U/g, and an incubation temperature of 48 °C. The resulting hydrolyzed corn starch films exhibited improved mechanical properties, including greater elasticity, strength, and stiffness, as well as higher values for water absorption index, transparency, and contact angle. The hydrolyzed corn starch films exhibited better energy dissipation properties than native corn starch films, making them suitable for applications requiring shock absorption or impact resistance. These results were confirmed by various analyzes, including FTIR, DSC, DMA, and AFM. The results demonstrated the potential of enzymatic hydrolysis to improve the mechanical properties and energy dissipation of corn starch films, expanding their potential applications in various industries.

## Figures and Tables

**Figure 1 polymers-15-01899-f001:**
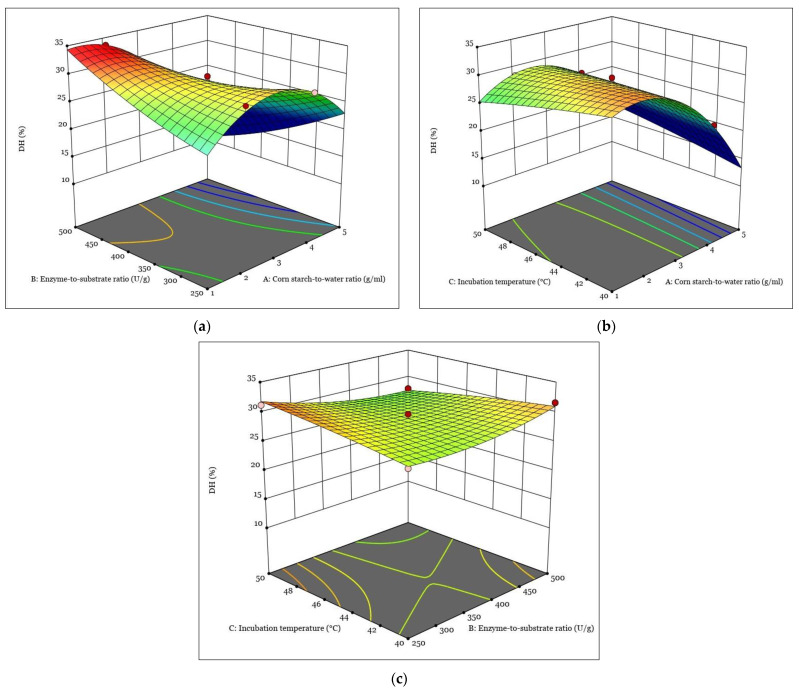
The response surfaces plots of the quadratic and 2FI polynomial models representing the influence of the interaction between independent variables on the DH: (**a**) interaction between A and B; (**b**) interaction between A and C; (**c**) interaction between B and C.

**Figure 2 polymers-15-01899-f002:**
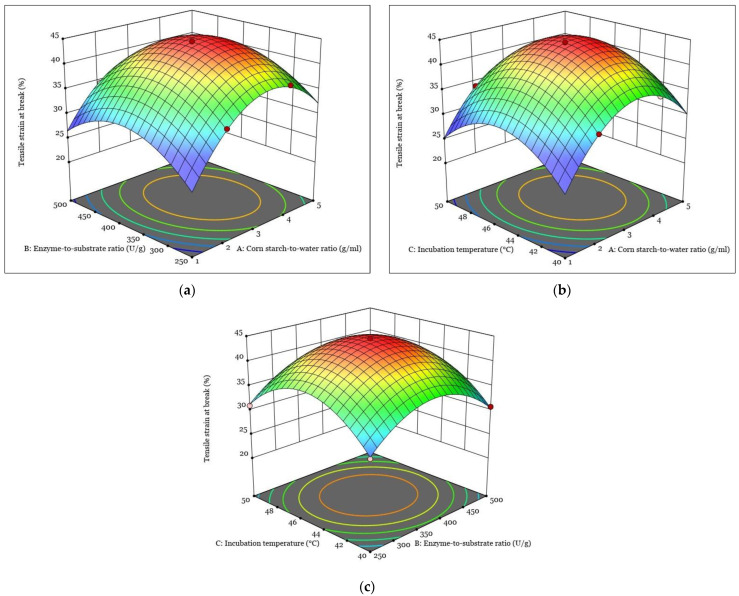
The response surfaces plots of the quadratic and 2FI polynomial models representing the influence of the interaction between independent variables on the tensile strain at break: (**a**) interaction between A and B; (**b**) interaction between A and C; (**c**) interaction between B and C.

**Figure 3 polymers-15-01899-f003:**
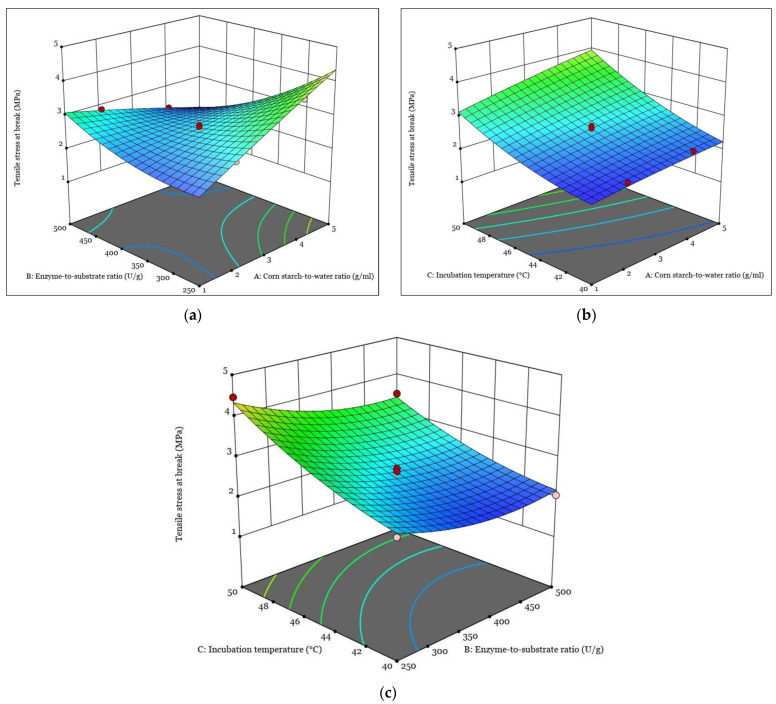
The response surfaces plots of the quadratic and 2FI polynomial models representing the influence of the interaction between independent variables on the tensile stress at break: (**a**) interaction between A and B; (**b**) interaction between A and C; (**c**) interaction between B and C.

**Figure 4 polymers-15-01899-f004:**
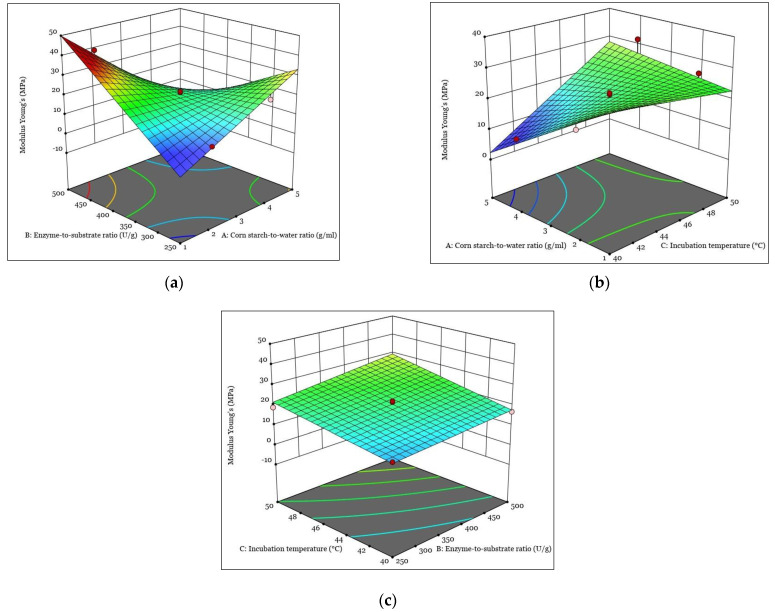
The response surfaces plots of the quadratic and 2FI polynomial models representing the influence of the interaction between independent variables on Young’s modulus: (**a**) interaction between A and B; (**b**) interaction between A and C; (**c**) interaction between B and C.

**Figure 5 polymers-15-01899-f005:**
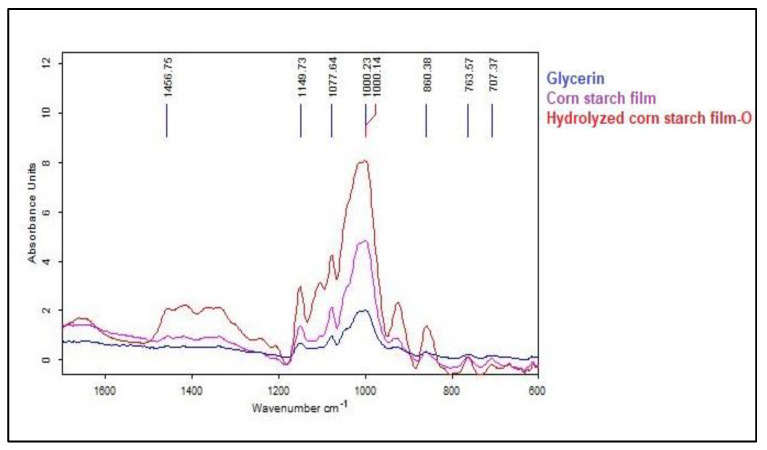
FTIR spectra of hydrolyzed corn starch (under optimum conditions) films samples and corn starch films.

**Figure 6 polymers-15-01899-f006:**
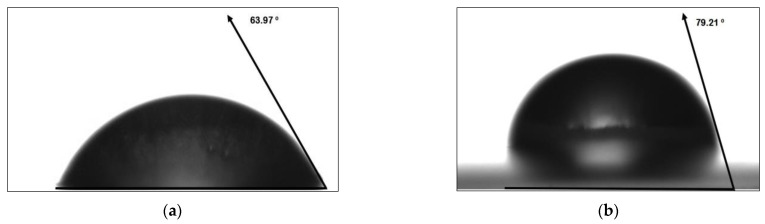
The contact angle of water droplets on starch-based films: (**a**) corn starch; (**b**) hydrolyzed corn starch.

**Figure 7 polymers-15-01899-f007:**
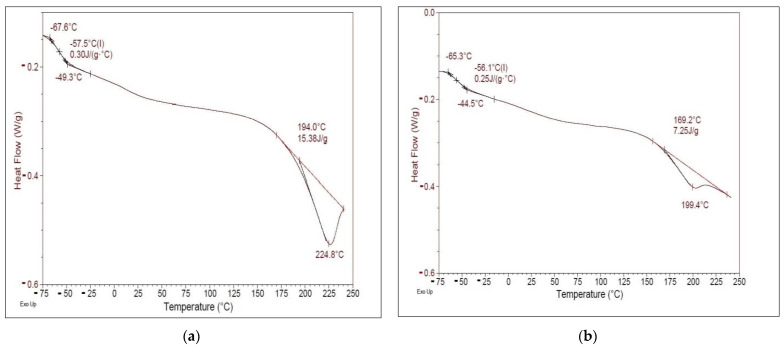
DSC thermograms of starch-based films: (**a**) native corn starch; (**b**) hydrolyzed corn starch under optimal conditions.

**Figure 8 polymers-15-01899-f008:**
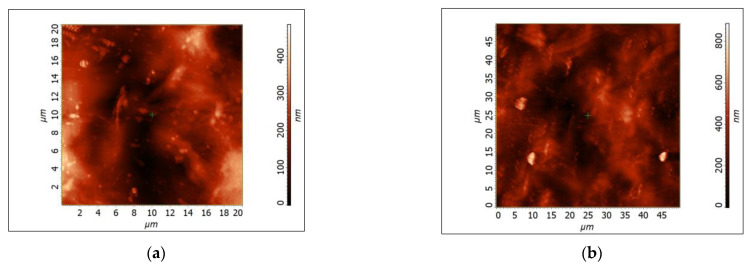
AFM 2D images of the hydrolyzed corn starch films: (**a**) 20 × 20 µm^2^; (**b**) 50 × 50 µm^2^.

**Figure 9 polymers-15-01899-f009:**
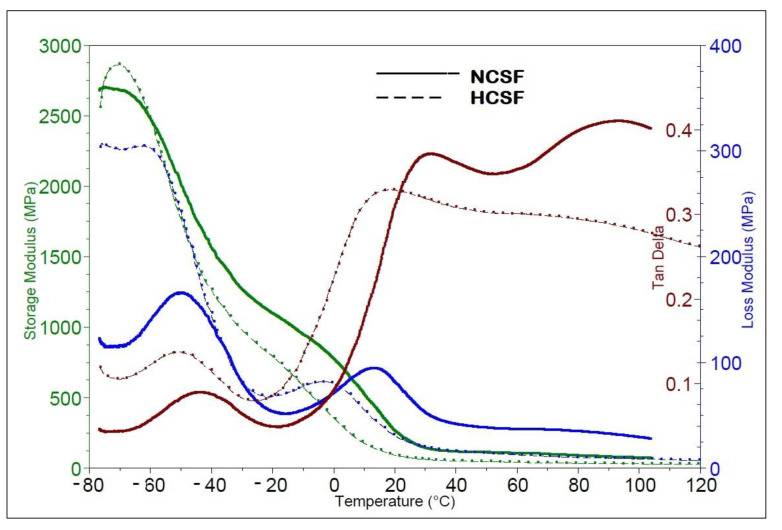
Storage modulus and tan δ plots for the native corn starch film (NCSF) and hydrolyzed corn starch film (HCSF).

**Table 1 polymers-15-01899-t001:** Actual values at the coded factor levels of the independent variables used in the RSM method.

Symbol	Independent Variable	Actual Values at Coded Factor Levels		
		−1	0	1
A	Corn-starch-to-water ratio (g/mL)	1:2	1:3	1:4
B	Enzyme-to-substrate ratio (U/g)	250	375	500
C	Incubation temperature (℃)	40	45	50

**Table 2 polymers-15-01899-t002:** BBD of the different formulations for the enzymatic hydrolysis conditions.

Runs	(A) Corn-Starch-to-Water Ratio g/mL	(B) Enzyme-to-Substrate Ratio U/g	(C) Incubation Temperature ℃
1	4(1)	375(0)	50(1)
2	2(−1)	250(−1)	45(0)
3	3(0)	250(−1)	40(−1)
4	3(0)	375(0)	45(0)
5	2(−1)	375(0)	40(−1)
6	3(0)	500(1)	50(1)
7	4(1)	500(1)	45(0)
8	4(1)	250(−1)	45(0)
9	3(0)	500(1)	40(−1)
10	3(0)	250(−1)	50(1)
11	2(−1)	500(1)	45(0)
12	3(0)	375(0)	45(0)
13	4(1)	375(0)	40(−1)
14	3(0)	375(0)	45(0)
15	2(−1)	375(0)	50(1)

**Table 3 polymers-15-01899-t003:** The values for experimental (actual), predicted, and residual for each response.

Run	DH (%)			Tensile Strain at Break (%)			Tensile Stress at Break (MPa)			Young’s Modulus (MPa)		
	Actual	Predicted	Residuals	Actual	Predicted	Residuals	Actual	Predicted	Residuals	Actual	Predicted	Residuals
1	25.59	24.97	0.6212	37.08	37.17	−0.0938	3.68	3.78	−0.0987	31.65	26.92	4.73
2	29.95	29.01	0.9375	33.53	33.30	0.2337	2.72	2.72	−0.0012	10.53	9.86	0.6671
3	28.41	28.73	−0.3163	29.55	29.88	−0.3275	2.54	2.64	−0.0975	14.44	13.8	0.6358
4	29.68	28.95	0.7267	44.21	44.39	−0.1800	2.73	2.64	0.0867	21.65	20.64	1.01
5	30.65	31.27	−0.6213	32.73	32.64	0.0937	2.14	2.04	0.0988	20.65	22.45	−1.8
6	27.85	27.53	0.3163	32.17	31.84	0.3275	3.50	3.40	0.0975	25.51	29.52	−4.01
7	20.95	21.89	−0.9375	36.84	37.07	−0.2338	2.31	2.31	0.0013	9.22	10.44	−1.22
8	28.53	28.54	−0.0050	37.72	37.36	0.3637	3.79	3.82	−0.0312	22.85	25.5	−2.65
9	31.59	30.97	0.6162	30.80	30.53	0.2700	2.06	2.19	−0.1300	16.86	17.66	−0.8017
10	31.18	31.80	−0.6163	31.02	31.29	−0.2700	4.47	4.34	0.1300	18.99	21.56	−2.57
11	33.65	33.65	0.0050	34.42	34.78	−0.3638	2.88	2.85	0.0312	38.85	36.75	2.1
12	28.68	28.95	−0.2733	44.45	44.39	0.0600	2.55	2.64	−0.0933	20.11	20.64	−0.5267
13	23.15	22.83	0.3212	35.75	35.79	−0.0363	2.32	2.19	0.1287	10.54	9.02	1.52
14	28.50	28.95	−0.4533	44.51	44.39	0.1200	2.65	2.64	0.0067	22.13	20.64	1.49
15	28.44	28.76	−0.3212	34.01	33.97	0.0362	3.24	3.37	−0.1287	25.57	24.16	1.41

**Table 4 polymers-15-01899-t004:** ANOVA of the regression coefficients of the fitted quadratic equations for the responses.

Variable	DH (%)			Tensile Strain at Break (%)			Tensile Stress at Break (MPa)			Young’s Modulus		
	Mean Square	F-Value	*p*-Value	Mean Square	F-Value	*p*-Value	Mean Square	F-Value	*p*-Value	Mean Square	F-Value	*p*-Value
Type of model	Quadratic			Quadratic			Quadratic			2FI		
Model	15.71	17.44	0.0029	38.51	239.24	<0.0001	0.729	29.47	0.0008	138.13	15.66	0.0005
Main effect												
A	74.85	83.09	0.0003	20.16	125.25	<0.0001	0.1568	6.34	0.0533	56.92	6.45	0.0347
B	2.03	2.25	0.1936	0.726	4.51	0.0871	0.9591	38.77	0.0016	69.8	7.91	0.0227
C	0.0684	0.076	0.7938	3.71	23.07	0.0049	4.25	171.74	<0.0001	192.37	21.81	0.0016
Interaction effect												
AB	31.81	35.31	0.0019	0.7832	4.87	0.0785	0.6724	27.18	0.0034	439.95	49.88	0.0001
AC	5.41	6	0.058	0.0006	0.0039	0.9527	0.0169	0.6832	0.4461	65.53	7.43	0.026
BC	10.6	11.76	0.0186	0.0025	0.0155	0.9057	0.06	2.43	0.18	4.2	0.4765	0.5096
Quadratic effect												
A^2^	11.2	12.43	0.0168	20.87	129.66	<0.0001	0.0002	0.0094	0.9267			
B^2^	4.14	4.59	0.085	150.53	935.16	<0.0001	0.3096	12.52	0.0166			
C^2^	0.2385	0.2648	0.6288	187.18	1162.86	<0.0001	0.1622	6.56	0.0506			
Lack of fit	1.23	3.05	0.2567	0.2515	9.98	0.0924	0.0358	4.4	0.1906	11.39	10.23	0.0917
R^2^	0.9691			0.9977			0.9815			0.9215		
Adjusted R^2^	0.9136			0.9935			0.9482			0.8627		

A = coded value of corn-starch-to-water ratio (g/mL), B = coded value of enzyme-to-substrate ratio (U/g), and C = coded value of incubation temperature (°C).

**Table 5 polymers-15-01899-t005:** Second-order polynomial equations for the models and the regression coefficients.

Response	Second-Order Polynomial Model
DH (%)	Actual equation: Y3 = 28.95 − 3.06A − 2.82AB − 1.63BC − 1.74A^2^
Tensile strain at break (%)	Coded equation: Tensile strain at break = 44.39 + 1.59A + 0.6813C − 2.38A^2^ − 6.38B^2^ − 7.12C^2^
Tensile stress at break (MPa)	Coded equation: Tensile stress at break = 2.64 − 0.3468B + 0.7288C − 0.41AB + 0.2896B^2^
Young’s modulus (MPa)	Coded equation: Young’s modulus = 20.64 − A + 2.95B + 4.90C − 10.49AB + 4.04AC

A = coded value of corn-starch-to-water ratio (g/mL), B = coded value of enzyme-to-substrate ratio (U/g), and C = coded value of incubation temperature (°C).

**Table 6 polymers-15-01899-t006:** Optimum conditions, experimental and predicted values of responses.

Optimum Conditions	Actual Levels	
Corn-starch-to-water ratio (g/mL)	1:2.8	
Enzyme-to-substrate ratio (U/g)	350	
Incubation temperature (°C)	48	
Response	Optimal predicted values	Optimal experimental values
DH (%)	29.45 ^a^	28.05 ± 0.2700 ^a^
Tensile strain at break (%)	47.54 ^b^	51.21 ± 0.3821 ^b^
Tensile stress at break (MPa)	8.31 ^c^	8.71 ± 0.4552 ^c^
Young’s modulus (MPa)	45.6	48.04 ± 0.4701

All results for the experimental values were expressed as mean value ± standard deviation of at least three measurements (n = 3). A statistically significant difference is denoted by different lowercase letters in the same column if *p* < 0.05. For all analyses, there were no statistically significant differences (*p* < 0.001) between groups.

**Table 7 polymers-15-01899-t007:** The roughness parameters of starch-based films.

Roughness Parameters	20 × 20 µm^2^	50 × 50 µm^2^
Sq (nm)	81.66	115.24
Sa (nm)	66.76	87.05

**Table 8 polymers-15-01899-t008:** DMA analysis results.

Sample	Step Transition, Storage Modulus, E′			Loss modulus, E″		Tan Delta	
	Onset	Midpoint (I)	End	Temperature	E″	Temperature	Tan δ
	°C	°C	°C	°C	MPa	°C	
NCSF_1_	−62.11	−49.57	−37.72	−50.61	164.9	−43.35	0.08889
NCSF_2_	0.32	9.58	24.35	13.14	93.88	31.7	0.3705
HCSF_1_	−63.66	−53.3	−44.35	−61.58	304.6	−51.09	0.1372
HCSF_2_	−13.51	−9.1	12.14	−3.48	81.74	17.82	0.3291

## Data Availability

No new data were created or analyzed in this study. Data sharing does not apply to this article.

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
