# Peer review of "Enhancing the Mechanical Properties of Corn Starch Films for Sustainable Food Packaging by Optimizing Enzymatic Hydrolysis"

_polymers, 2023, doi:10.3390/polym15081899_

Round 1

Reviewer 1 Report

Abstract: Mention the enzyme name in abstract. What was the composition of enzymatically hydrolyzed starch?? Why better mechanical properties were associated with hydrolysis?

Introduction: How enzymatic hydrolysis of starch is an important process for the development of biodegradable films (Line 68)??

Please include structure and functional changes expected in starch after hydrolysis.

Materials and method: How enzyme proportion was calculated (250-500 U/g) to hydrolyze the starch?

Quantification of reducing sugars should be done by DNS standard method using glucose as reference standard. The current method used seems to be not very reproducible.

What was the effect of starch hydrolysis on water or oxygen permeability??

3D RSM graphs can be put in supplementary file and only important influence parameters should be briefly discussed.

For FTIR high quality file should be provided.

Conclusion needs to be concrete and very focus, rather than general discussion of results.

Author Response

Dear reviewer,

We would like to thank the reviewer for the valuable comments/suggestions/recommendations that helped us improve our work.

Best regards,

Reviewer 2 Report

- 46-49 please also provide some other natural polymers in addition to starch too, please see some examples at http://ojs.kmutnb.ac.th/index.php/ijst/article/view/5183/3616, http://ojs.kmutnb.ac.th/index.php/ijst/article/view/5408/3735

- 83, how, please explain and add reference

-133-134, please indicate the solid loading ratio

- 144, why not taking from the 4th time?

- 285-286 what's the criteria to justify that the experimental values and anticipated values are well agreed.

- 287 please indicate the p-value. and is it table 4?

- fig 5 please discuss why several peaks found in the hydrolysed corn starch film are not found in other samples. for example 1456, ~1100, ~900

-fig 6 it's better to add line to show the angle of each sample

minor check for spelling

Author Response

(The authors gave the same response as above.)

Round 2

Reviewer 1 Report

The manuscript is revised as advised.